# Nucleolar fibrillarin is an evolutionarily conserved regulator of bacterial pathogen resistance

Varnesh Tiku[1,2,4], Chun Kew[1], Parul Mehrotra[1,5], Raja Ganesan[2,3], Nirmal Robinson[2,6] & Adam Antebi[1,2]

Innate immunity is the first line of defense against infections. Pathways regulating innate responses can also modulate other processes, including stress resistance and longevity. Increasing evidence suggests a role for the nucleolus in regulating cellular processes implicated in health and disease. Here we show the highly conserved nucleolar protein, fibrillarin, is a vital factor regulating pathogen resistance. Fibrillarin knockdown enhances resistance in *C. elegans* against bacterial pathogens, higher levels of fibrillarin induce susceptibility to infection. Pathogenic infection reduces nucleolar size, ribsosomal RNA, and fibrillarin levels. Genetic epistasis reveals fibrillarin functions independently of the major innate immunity mediators, suggesting novel mechanisms of pathogen resistance. Bacterial infection also reduces nucleolar size and fibrillarin levels in mammalian cells. Fibrillarin knockdown prior to infection increases intracellular bacterial clearance, reduces inflammation, and enhances cell survival. Collectively, these findings reveal an evolutionarily conserved role of fibrillarin in infection resistance and suggest the nucleolus as a focal point in innate immune responses.

[1] Max Planck Institute for Biology of Ageing, Joseph Stelzmann Strasse 9b, 50931 Cologne, Germany. [2] Cologne Excellence Cluster on Cellular Stress Responses in Aging-Associated Diseases (CECAD), University of Cologne, 50674 Cologne, Germany. [3] Institute for Medical Microbiology, Immunology and Hygiene, University of Cologne, 50674 Cologne, Germany. [4] Present address: Department of Infectious Diseases, Genentech Inc., 1 DNA Way, South San Francisco, CA 94080, USA. [5] Present address: VIB-Center for Inflammation Research, VIB - Ghent University, Technologiepark 927, 9052 Ghent, Belgium. [6] Present address: Centre for Cancer Biology, University of South Australia, HB11-35 UniSA CRI Building, North Terrace, 5001 Adelaide, Australia. These authors contributed equally: Varnesh Tiku, Chun Kew, Parul Mehrotra. These authors jointly supervised this work: Nirmal Robinson, Adam Antebi. Correspondence and requests for materials should be addressed to N.R. (email: nirmal.robinson@uk-koeln.de) or to A.A. (email: Antebi@age.mpg.de)

nnate immunity helps combat infections in the absence of prior exposure to pathogens. Pathogen recognition by the host entails detection of characteristic molecular patterns associated with the pathogen, so-called pathogen-associated molecular patterns (PAMPs)[1]. The host uses specialized factors called pattern recognition receptors (PRRs) to detect PAMPs triggering an immune response, which involves upregulation of immune response genes. However, PAMPs are associated with a wide range of microbes, both pathogenic and commensal. Therefore, the question arises how does an organism distinguish between a pathogenic and a non-pathogenic microbe if it is only reliant on the PAMPs detection system.

A relatively new concept in the field of innate immunity is the so-called effector-triggered immunity, whereby host cells are alerted to the pathogen by the associated damage caused by pathogen secreted toxins or virulence factors[2,3]. For instance, virulent toxins produced by diverse pathogens including *Pseudomonas aeruginosa*[4], *Legionella pneumophila*[5], and Shiga toxin-secreting *Escherichia coli*[6,7] often suppress host messenger RNA (mRNA) translation to inhibit expression of anti-microbial factors and thereby assist in bacterial survival[8]. Many of these toxins have different conformational properties and it would be inefficient for the host to have specific PRRs capable of recognizing all these diverse structures. Therefore, nature has employed a rather efficient way of pathogen detection by encountering the damage caused by the pathogen to the host. This theme of effector-triggered immunity has been well established in plants[9,10] and more recently has been appreciated in animal models including *Caenorhabditis elegans*[11].

The nucleolus is best known as a sub-nuclear organelle where ribosomal RNA (rRNA) is synthesized and assembled into ribosomal subunits. Recent evidence, however, also suggests that the nucleolus aids in the assembly of other ribonucleoprotein particles, including splicing factors, signal recognition particle, small interfering RNA (siRNA) machinery, stress granules, and telomerase[12], and regulates physiologic processes such as stress responses[13] and aging[14,15]. Interestingly, a handful of studies have linked the nucleolus with viral infection. Several viral proteins localize in the nucleolus after infection[16], and viruses subvert the host cell by recruiting nucleolar proteins for viral replication[16]. However, the potential of this organelle in innate immunity in bacterial infections remains largely unexplored.

Here we report that downregulation of the highly conserved nucleolar protein fibrillarin increases infection resistance of *C. elegans* against the bacterial pathogens *Staphylococcus aureus*, *Enterococcus faecalis*, and *P. aeruginosa*. We show that bacterial infection leads to a reduction in nucleolar size, rRNA, and fibrillarin levels. Fibrillarin knockdown confers resistance to the known infection-sensitive mutants, suggesting that fibrillarin reduction-mediated protection is independent of the major immune response pathways in *C. elegans*. Fibrillarin levels are also reduced upon infection in mammalian cells and its prior knockdown enhanced clearance of intracellular bacteria, improved cell survival, and reduced inflammation after infection. We propose that fibrillarin acts as a central node in a regulatory network engaged in imparting immunity against bacterial pathogens, conserved across evolution.

## Results

### Fibrillarin regulates resistance to bacterial infection.
*C. elegans* *fib-1* encodes the highly conserved nucleolar methyltransferase fibrillarin, which is a vital factor in the C/D small nucleolar ribonucleoprotein (snoRNP) complex. Fibrillarin mediates 2′-O-ribose methylation of rRNA thereby assisting in the maturation of rRNA[17,18], and also methylates histone H2AGln105 at the rDNA

locus[19]. We recently reported that *fib-1* is downregulated in multiple well-established longevity mutants of *C. elegans* and that *fib-1* knockdown reduces nucleolar size and extends lifespan in worms[14]. Since genes that promote lifespan extension often induce tolerance against multiple stress conditions including pathogenic infections[20–22], we wondered if *fib-1* reduction could confer infection resistance against pathogens. We knocked-down *fib-1* using RNAi and monitored infection resistance in worms. Since *fib-1* is an essential gene, we resorted to RNAi from larval stage 3 (L3) up to day one of adulthood (around 30 h of *fib-1* RNAi), which led to a significant reduction in FIB-1 levels without causing any developmental defects (Supplementary Fig. 1A). Interestingly, animals with *fib-1* RNAi displayed significantly increased survival upon infection with pathogens *S. aureus*, *E. faecalis*, and *P. aeruginosa* (Fig. 1a, b and Supplementary Fig 1B). *fib-1* knockdown did not affect other stress responses including heat, cold, and oxidative stress resistance, suggesting that *fib-1* specifically regulates pathogen resistance (Supplementary Fig 1C–E). *fib-1* RNAi did not affect pharyngeal pumping rate, ruling out the possibility of differences in bacterial intake (Supplementary Fig. 1F). Reduction in major cellular processes is known to induce food aversion behavior in *C. elegans*, which has been linked to innate immunity[23]. However, *fib-1* knockdown-mediated improvement in infection resistance was independent of food aversion behavior (Supplementary Fig. 1G, H). We and others have shown previously the B-box protein NCL-1/TRIM2 to be an upstream negative regulator of FIB-1[14,24,25]. FIB-1 levels are highly upregulated in *ncl-1* loss-of-function mutants. Therefore, we tested the survival of *ncl-1* mutants upon infection. Interestingly, we found *ncl-1* mutants were more susceptible to infection, suggesting that increased levels of *fib-1* are detrimental for survival upon infection challenge (Fig. 1c, d). Taken together, our results reveal a state of protection conferred by *fib-1* reduction that helps worms survive longer upon infection.

### Fibrillarin and nucleolar size are reduced upon infection.
Next, we assessed the levels of FIB-1 after infection. We performed western blot to detect the endogenous levels of FIB-1 after 12-h infection in wild-type and more susceptible *ncl-1* mutants. We observed a downregulation of FIB-1 protein levels in wild-type worms after infection with *S. aureus*, *E. faecalis*, and *P. aeruginosa* (Fig. 2a, b, Supplementary Fig. 2A). *ncl-1* mutants also exhibited a slight reduction of FIB-1 after infection, but the levels remained significantly higher compared to wild type (Fig. 2a, b). We also obtained similar results with the FIB-1::GFP strain harboring a translational fusion; the GFP signal was significantly downregulated after infection with *S. aureus* (Fig. 2c). Notably, we did not observe significant transcriptional changes of *fib-1* using qPCR, suggesting a post-transcriptional response (Supplementary Fig. 2B). In our previous study, we reported that the nucleolar size decreased in worms subjected to *fib-1* RNAi[14]. Because we observed a reduction in FIB-1 levels after infection, we wondered if nucleolar size changes correspondingly. Indeed, we observed a significant decrease (~25%) in the nucleolar size of worms after 12-h infection with *S. aureus* and *E. faecalis* (Fig. 2d, e). The size of nucleoli did not change when worms were fed heat-killed *S. aureus* and *E. faecalis*, suggesting that the nucleolar size reduction was caused by active infection (Supplementary Fig. 2C, D). Since the nucleolus is the site of rRNA maturation and FIB-1 plays a crucial role in this process, we wondered if rRNA levels were altered after infection. As predicted, infection with *S. aureus* led to a reduction of mature rRNA levels (Fig. 2f). *ncl-1(+)* is known to limit nucleolar size in worms; *ncl-1* null mutants possess enlarged nucleoli in multiple tissues[26,27]. We assessed the

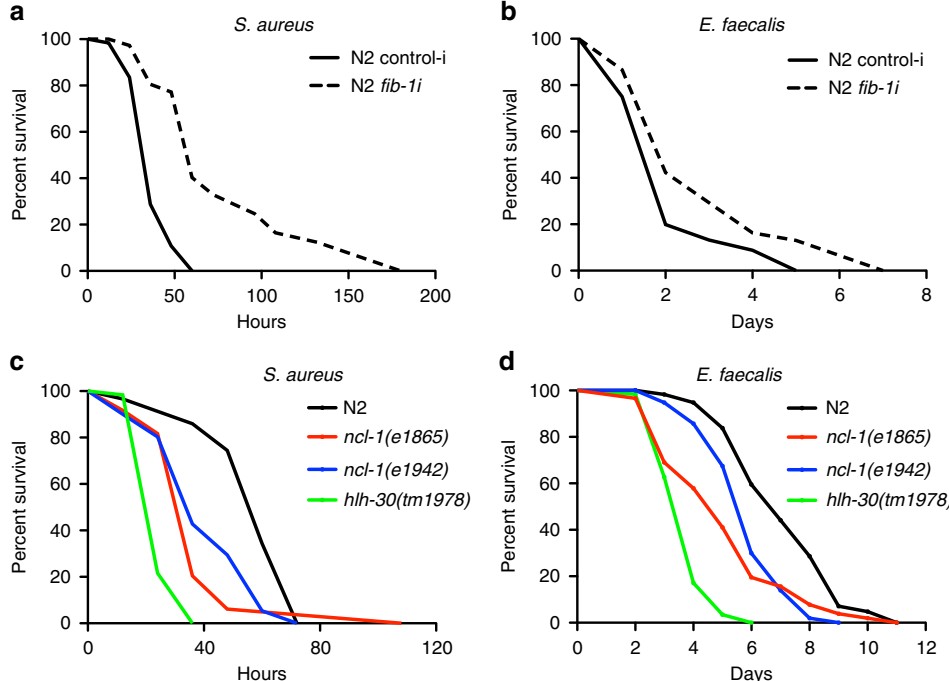

**Fig. 1** *fib-1*/fibrillarin regulates bacterial infection resistance in *C. elegans*. **a**, **b** *fib-1* knockdown improves survival of wild-type N2 worms upon *S. aureus* and *E. faecalis* infection ($P < 0.0001$). **c**, **d** *ncl-1* mutants (*e1865* and *e1942*) are short-lived compared to wild-type N2 upon infection with *S. aureus* and *E. faecalis* ($P < 0.0001$). *hlh-30(tm1978)* served as a control for infection. Survival experiments were performed three times independently. *P*-values were calculated by log-rank test

nucleolar size of *ncl-1* mutants after infection and observed that unlike wild-type worms, nucleolar size remained enlarged (Fig. 2d, e). Taken together, these results suggest that a reduction in FIB-1 levels and nucleolar size might be a host response toward combating infection challenge. *ncl-1* mutants are somewhat refractory in this response, which might explain their increased susceptibility.

**Fibrillarin affects survival of pathogen sensitive mutants**. We next sought to investigate the link between *fib-1* and established major defense–response pathways in *C. elegans*. We examined genetic epistasis between *fib-1* and known vital mediators of defense–response upon pathogenic insult in worms. p38 MAP kinase (MAPK) pathway is a key evolutionarily conserved defense–response pathway that is activated upon microbial infection and mediates important downstream transcriptional changes[28]. p38 MAPK, which is encoded by *pmk-1* in *C. elegans*, is the major regulator of this pathway[29]. A mutation in *pmk-1* leads to increased susceptibility of worms upon infection with diverse pathogens. *fib-1* knockdown significantly improved the survival of *pmk-1* (Fig. 3a, Supplementary Fig. 3A). Next we tested *hlh-30*/TFEB, a crucial factor promoting autophagy and anti-microbial gene expression as part of the host response upon infection in worms and mammals[30]. Previous reports have shown that HLH-30/TFEB is nuclear localized upon bacterial infection and is required for activation of autophagy and anti-microbial gene transcription in *C. elegans*[30,31]. However, *fib-1* knockdown did not affect HLH-30 nuclear localization (Supplementary Fig. 4A, B). Interestingly, knocking down *fib-1* in *hlh-30* mutants also led to a significant increase in survival of these animals upon infection (Fig. 3b, Supplementary Fig. 3A). *fib-1* RNAi did not affect autophagic flux under uninfected conditions, as measured by the cleavage of the autophagosomal membrane protein LC3/ LGG-1, although under starvation and infection autophagic flux increased as expected (Supplementary Fig. 4C). These findings

suggest that the protective effects imparted by *fib-1* reduction function independently of HLH-30/TFEB. Another proteostasis mechanism, the ubiquitin-proteasome system (UPS) was also examined upon *fib-1* knockdown and *S. aureus* infection. Total levels of ubiquitinated proteins remained similar in control and *fib-1* RNAi animals and there was also no noticeable difference upon infection, suggesting that the enhanced survival conferred by *fib-1* RNAi does not obviously arise from increased proteolytic mechanisms (Supplementary Fig 4D). Finally we examined the epistasis with *daf-16*/FOXO, another important highly conserved transcription factor driving anti-microbial genes[32,33] and infection resistance of *daf-2* mutants[21]. Similar to the effects on *pmk-1* and *hlh-30*, *daf-16* showed increased survival on infection upon *fib-1* RNAi (Fig. 3c, Supplementary Fig. 3A). Furthermore, reduced survival of *ncl-1* mutants was also rescued with *fib-1* RNAi (Fig. 3d, Supplementary Fig. 3B, C). Taken together, our results reveal FIB-1 as a novel regulator of host response toward infection working downstream or independently of established defense–response pathways in worms.

**Fibrillarin reduction induces translation suppression**. To investigate the mechanism behind *fib-1* reduction-mediated pathogen resistance, we studied the involvement of mRNA translation. Recent studies have reported that worms detect translation suppression by infection as a means to activate defense response[34,35]. Since FIB-1 is a methyltransferase involved in rRNA maturation and ribosome biogenesis, we hypothesized that *fib-1* RNAi might lead to a reduction in ribosome biogenesis and thereby translation, thus activating the defense response. To test this possibility, we examined a well-established reporter gene, *irg-1* (infection response gene-1) fused to GFP. *irg-1* is activated by the bZIP transcription factor ZIP-2 during pathogen challenge. In response to a block in translation upon infection with *P. aeruginosa*, ZIP-2 itself is preferentially translated in a manner dependent upon its upstream 5′-UTR[36]. After infection with

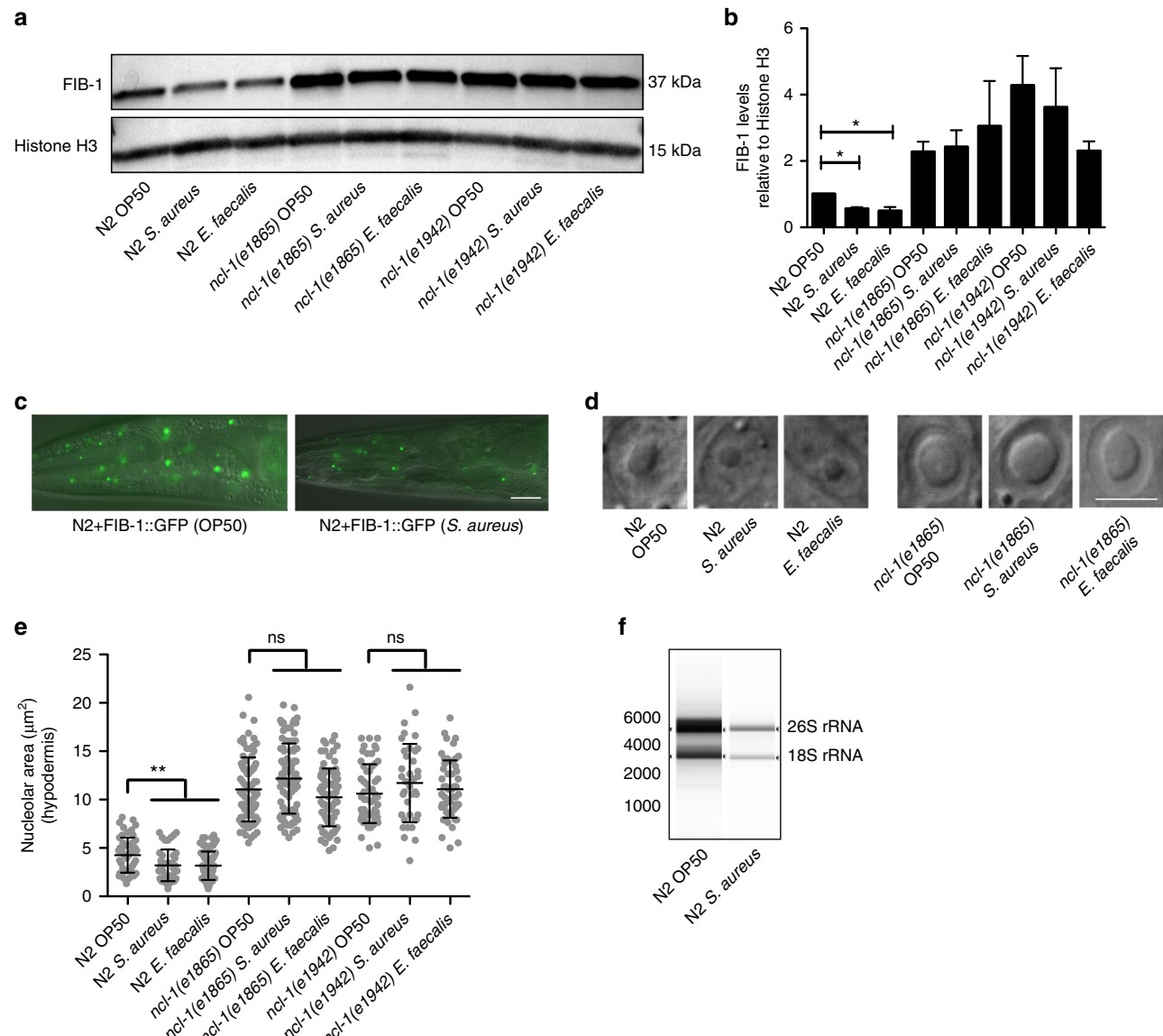

**Fig. 2** FIB-1/fibrillarin and nucleolar size are reduced upon bacterial infection. **a**, **b** FIB-1 levels are significantly reduced in wild-type N2 after *S. aureus* and *E. faecalis* infection. FIB-1 is modestly reduced in *ncl-1* mutants after infection but the levels are still higher relative to wild-type N2 after infection. Error bars represent mean ± s.e.m. of three independent biological replicates. **c** FIB-1::GFP shows reduced fluorescent signal after infection with *S. aureus*. **d**, **e** Nucleolar size in hypodermal cells of wild-type N2 worms is reduced after 12-h infection with *S. aureus* and *E. faecalis*. Nucleolar size in hypodermal cells of *ncl-1* mutants after infection remains unaffected. Error bars represent mean ± s.d. **f** *S. aureus* infection for 12 h reduces 26S and 18S rRNA levels in wild-type *C. elegans*. RNA extracted from equal number of uninfected and infected worms was analyzed using a bioanalyzer. Scale bars represent 20 (**c**) and 5 μm (**d**). *P < 0.05, **P < 0.01, ns—non-significant, unpaired *t*-test

*S. aureus*, we observed a strong induction of *irg-1* (Fig. 4a, b). Interestingly, we observed a similar induction of *irg-1* transgene as well as the *irg-1* transcript upon *fib-1* knockdown without infection, suggesting that *fib-1* RNAi might reduce translation and thereby activate *irg-1* (Fig. 4c–e). As a control, the translational inhibitor cycloheximide also induced *irg-1* in a similar manner (Supplementary Fig. 5A). This was not a generalized inflammatory response, however, as other major infection related genes did not change upon *fib-1* RNAi (Supplementary Fig. 5B). Consistent with reduced translation, *fib-1* RNAi led to a reduction in mature rRNA levels (Fig. 4f). Additionally, *fib-1* knockdown also attenuated translation as suggested by a reduction in puromycin incorporation (Fig. 4g, h). We also observed an increase in the precursor-rRNA (pre-rRNA) levels upon *fib-1* knockdown (Supplementary Fig. 5C). Conceivably, this pre-rRNA species

accumulates due to a block in the maturation process by reduced FIB-1 levels. These data also suggest that the decrease of mature rRNA levels in *fib-1* knockdown animals occurs post-transcriptionally.

Since we observed effects of *fib-1* reduction on rRNA levels and translation, we next sought to test if *fib-1*-mediated protection is mediated by a reduction in translation. To address this question, we first asked if mutants with reduced translation have improved infection resistance. We used *ifg-1* and *ife-2* mutants, which are known to have reduced translation[37–39]. *ifg-1* encodes the translation initiation factor eIF4G, while *ife-2* encodes the cap binding initiation factor eIF4E. Indeed *ifg-1* and *ife-2* strains were significantly resistant to infection (Fig. 4i, Supplementary Fig. 5D). Moreover, *fib-1* RNAi only marginally enhanced the survival of *ifg-1* mutants upon infection (Fig. 4j, k). These results collectively

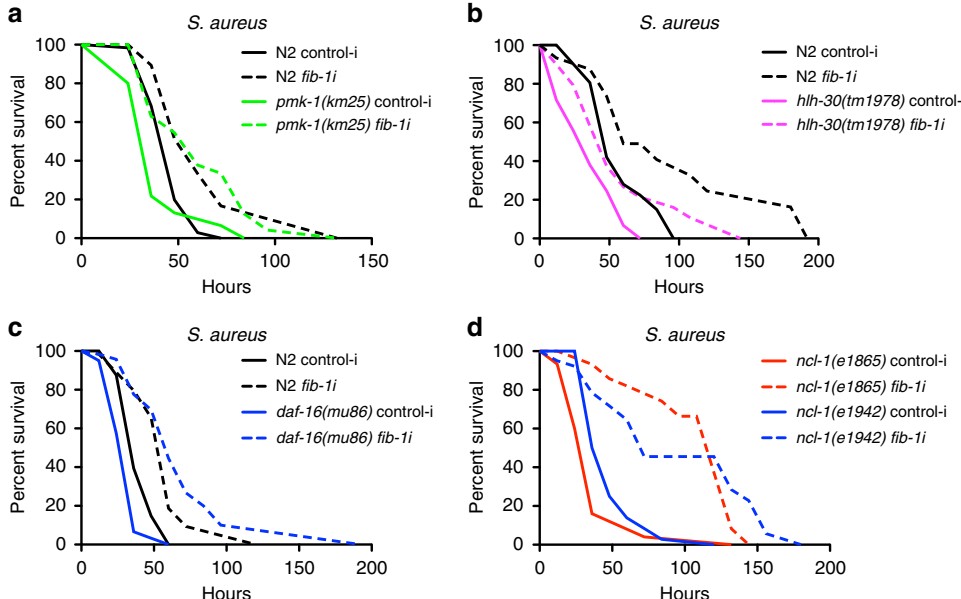

**Fig. 3** *fib-1*/fibrillarin reduction improves resistance of infection-sensitive mutants. **a**–**d** *fib-1* RNAi significantly improves the survival of infection-sensitive *pmk-1(km25)* (*P* = 0.0001), *hlh-30(tm1978)* (*P* = 0.0021), *daf-16(mu86)* (*P* < 0.0001) and *ncl-1(e1865* and *e1942*) mutants (*P* < 0.0001) upon *S. aureus* infection. Survival experiments were performed three times independently. *P*-values were calculated by log-rank test

suggest that *fib-1* knockdown-mediated pathogen resistance mechanistically overlaps with infection resistance conferred by translational reduction.

**Fibrillarin reduction protects mammalian cells**. We next sought to understand the potential role of fibrillarin in imparting immunity against infections in mammalian systems. To begin with, we wondered if mammalian fibrillarin levels are also perturbed after infection, as observed in *C. elegans*. We found that HeLa cells infected with *S. aureus* exhibited significantly reduced levels of fibrillarin at varying multiplicities of infections (MOIs) (Fig. 5a). Similarly mouse bone marrow-derived macrophages infected with *S. aureus*, *E. faecalis*, *Salmonella typhimurium*, and *Listeria monocytogenes* showed reduced levels of fibrillarin after 24 h of infection (Fig. 5b, c). Moreover, we also observed a modest reduction in nucleolar size in THP1 cells 24 h post infection with *S. aureus* similar to our results in worms, indicating that nucleolar size reduction is a conserved host response to infection (Fig. 5d, e).

We next wondered whether fibrillarin reduction post infection in mammalian systems is protective. We reasoned that if fibrillarin was reduced before infection, it could prime a host response to incoming pathogens. Using macrophages and HeLa cells, we investigated the role of fibrillarin in regulating infection resistance in the mammalian system. A major feature of *S. aureus* infection is the ability of the pathogen to induce inflammation and host-cell death, a phenomenon attributed to the pathogenicity of the bacteria[40]. We performed siRNA-mediated silencing of fibrillarin 48 h before infection in HeLa cells (Supplementary Fig. 6A) and compared the percentage of cells surviving at 6 and 24 h post infection in both fibrillarin and control siRNA-treated cells. Fibrillarin knockdown cells displayed significantly better survival after infection with *S. aureus* compared to control siRNA-treated cells as assayed by lactate dehydrogenase (LDH) cytotoxicity assay and trypan blue staining (Supplementary Fig. 6B, C). Murine bone marrow-derived macrophages also exhibited significantly reduced cytotoxicity after treatment with fibrillarin siRNA (Fig. 5f). Interestingly, fibrillarin knockdown improved clearance of intracellular *S. aureus* in murine bone

marrow-derived macrophages (Fig. 5g), which possibly explains the augmented resistance in these cells to infection upon fibrillarin knockdown. Conversely, fibrillarin overexpression modestly enhanced the susceptibility of HeLa cells to infection (Supplementary Fig. 6D). Fibrillarin knockdown and overexpression did not influence bacterial uptake by cells as measured by comparing intracellular colony forming units (CFU) with respective controls (Supplementary Fig. 6E, F) ruling out the possibility of differences in bacterial internalization with fibrillarin siRNA. Furthermore, fibrillarin knockdown prior to infection led to a reduction of pro-inflammatory cytokines interleukin (IL)-6 and IL-8 (Fig. 5h), and an induction of anti-inflammatory cytokine IL-10 (Fig. 5i) indicative of reduced inflammation. Immunofluorescence using GFP-labeled *S. aureus* showed increased apposition of the intracellular bacteria to lysosomes in cells treated with fibrillarin siRNA compared to control siRNA (Fig. 5j, k), which might explain increased intracellular bacterial clearance and enhanced cell survival upon fibrillarin knockdown. Taken together, these results point toward a conserved mechanism of bacterial infection resistance possibly imparted by increased phagosome maturation and reduced inflammation that is mediated by fibrillarin knockdown in mammalian cells.

## Discussion
Perturbation of biological systems by infection leads to a multi-layer complex cellular and organismal response. Whether this response ultimately leads to clearance of infection or collapse of the host system is largely dependent on the extent and nature of the cellular pathways perturbed and the interplay between the host and the pathogen.

Cellular organelles such as mitochondria, endoplasmic reticulum (ER), and lysosome have long been identified as signaling hubs that help manage infection. However, a role for the nucleolus in mediating the innate response to pathogenic stress is relatively unstudied. In this work, we identified the nucleolar protein fibrillarin as a novel player in regulating bacterial pathogen resistance of *C. elegans*. Fibrillarin reduction increases the survival of worms challenged with *S. aureus*, *E. faecalis*, and

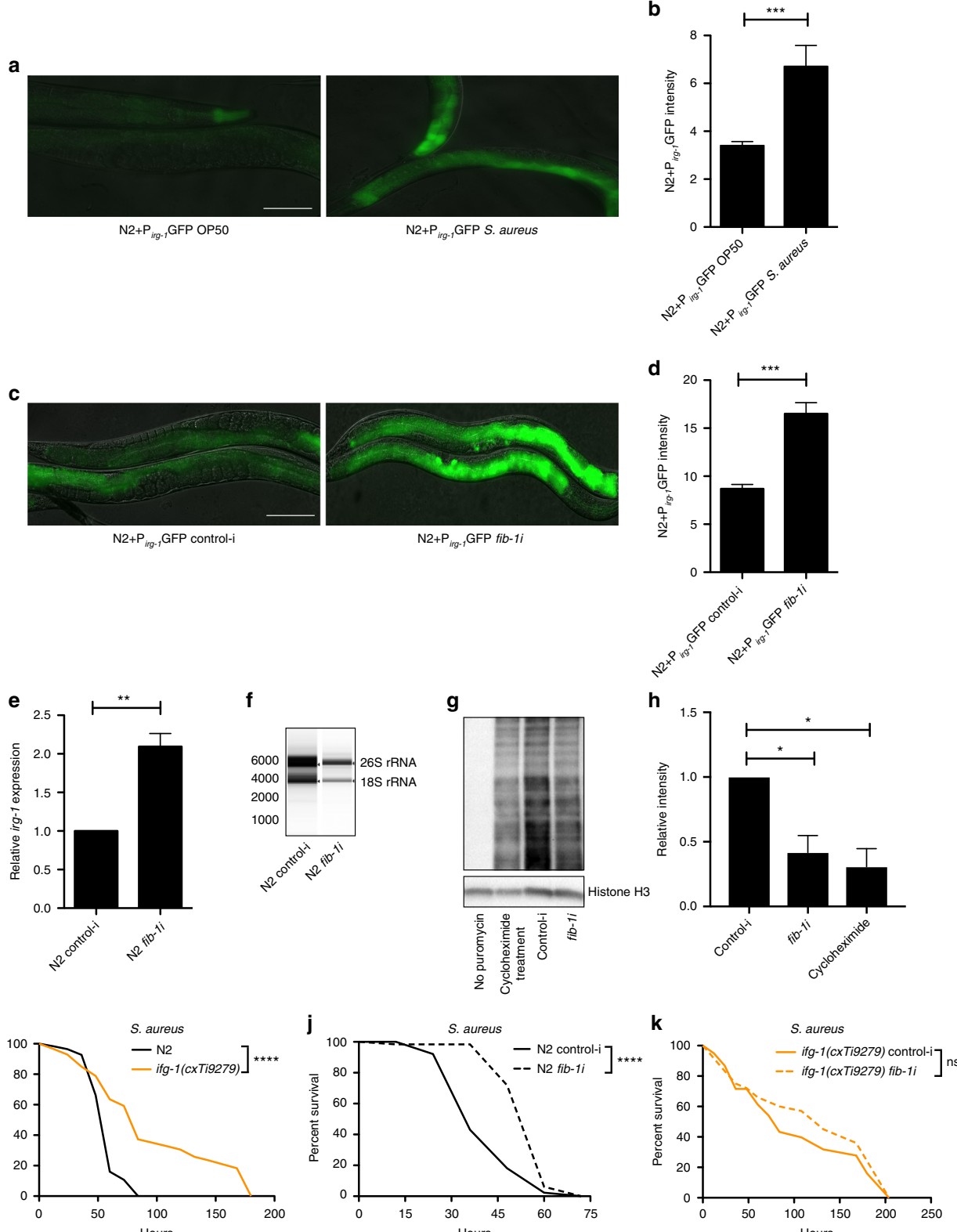

**Fig. 4** *fib-1*/fibrillarin reduction induces translation suppression. **a**, **b** Twelve-hour *S. aureus* infection induces P$_{irg-1}$GFP reporter. Error bars represent mean ± s.e.m. **c–e** *fib-1* knockdown induces P$_{irg-1}$GFP reporter and mRNA expression of *irg-1*. Error bars represent mean ± s.e.m. **f** *fib-1* RNAi reduces the levels of 26S and 18S rRNA in worms. RNA extracted from equal number of worms was analyzed using a bioanalyzer. **g**, **h** *fib-1* RNAi treatment reduced puromycin incorporation suggestive of reduced translation. No puromycin and cycloheximide treatments served as controls. Error bars represent mean ± s.e.m. **i** *ifg-1 (cxTi9279)* exhibits significantly extended survival compared to wild-type N2 upon *S. aureus* infection ($P < 0.0001$, log-rank test). **j**, **k** *fib-1* knockdown significantly increases the survival of wild-type N2 ($P < 0.0001$, log-rank test) but not of *ifg-1(cxTi9279)* ($P = 0.74$, log-rank test) upon *S. aureus* infection. Survival experiments were performed three times independently. Scale bars represent 100 μm. *$P < 0.05$, **$P < 0.01$, ***$P < 0.001$, ****$P < 0.0001$, ns—non-significant, unpaired *t*-test

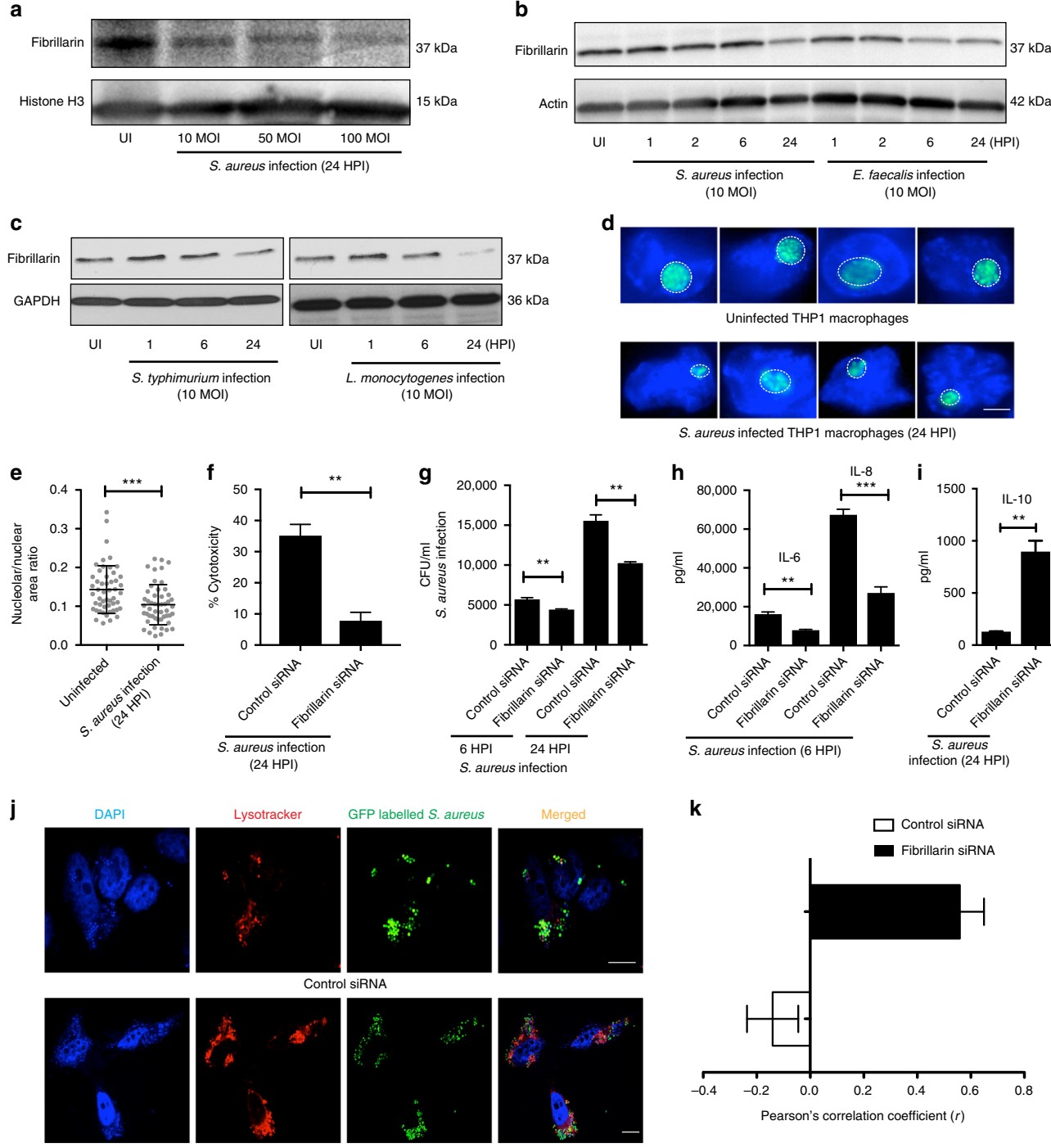

**Fig. 5** Fibrillarin reduction protects mammalian cells against bacterial pathogens. **a** *S. aureus* infection leads to a reduction of fibrillarin levels in HeLa cells. **b**, **c** Mouse bone marrow-derived macrophages show a reduction in fibrillarin 24 h post infection with *S. aureus*, *E. faecalis*, *S. typhimurium* and *L. monocytogenes*. **d**, **e** Twenty-four hours of *S. aureus* infection leads to a reduction in nucleolar size of THP1 macrophages. Error bars represent mean ± s.d. **f** Fibrillarin siRNA reduces cytotoxicity relative to control siRNA after 24 h of *S. aureus* infection (MOI 10) in murine bone marrow-derived macrophages. Error bars represent mean ± s.e.m., unpaired *t*-test. **g** Fibrillarin siRNA leads to a better clearance of intracellular pathogens thereby lowering the number of CFU per mL after 6 and 24 h of *S. aureus* infection (MOI 10) in murine bone marrow-derived macrophages. Error bars represent mean ± s.e.m., unpaired *t*-test. **h** Six hours after *S. aureus* infection, ELISA results show a decrease in pro-inflammatory cytokines interleukin 6 and 8 after fibrillarin siRNA treatment in HeLa cells. Error bars represent mean ± s.e.m., unpaired *t*-test. **i** ELISA results show an increase in anti-inflammatory cytokine interleukin 10, 24 h post infection with *S. aureus* upon fibrillarin knockdown in mouse bone marrow-derived macrophages. Error bars represent mean ± s.e.m, unpaired *t*-test. **j**, **k** HeLa cells infected with GFP-labeled *S. aureus* and stained with lysotracker show increased co-localization of bacteria with lysosomes in cells treated with fibrillarin siRNA. **P < 0.01, ***P < 0.001, unpaired *t*-test. Scale bars represent 4 (**d**) and 10 μm (**j**). UI uninfected, HPI hours post infection, MOI multiplicity of infection

*P. aeruginosa* infection and conversely *ncl-1*/TRIM2 mutants that possess higher levels of fibrillarin[14,24,25] are more susceptible to infection. Since *C. elegans* is largely dependent on innate immune signaling for pathogen resistance, the results indicate that higher levels of fibrillarin suppress innate immunity against bacterial pathogens. Interestingly, active bacterial infection and not heat-killed bacteria lead to a reduction in nucleolar size. This indicates that a reduction in nucleolar size and a decrease in fibrillarin levels constitute a host response mounted against live infection. *C. elegans* pathogen defense pathways are activated by a number of important factors including PMK-1/p38 MAPK, HLH-30/TFEB, and DAF-16/FOXO[41]. However, it still remains unclear how these different molecules coordinate downstream mechanisms to confer pathogen resistance. Our study suggests that fibrillarin regulates infection resistance as a convergent factor genetically downstream or parallel to these major players. How fibrillarin itself is regulated remains an interesting area of research to further pursue.

Pathogens often disrupt core cellular processes by delivering toxins so as to disable essential processes and pathways that would otherwise help mount a defense response. Multiple studies have reported that disruption of major cellular processes including mitochondrial respiration, proteasomal activity, microtubular dynamics, and mitochondrial unfolded protein response (UPR) can provoke immune-responsive genes, corroborating the notion of effector-triggered immunity[23,34,35,42]. Our work suggests a novel connection between the nucleolus and anti-bacterial immune function, and provides crucial evidence for a vital role of translation in imparting effector-triggered immunity. Improved infection resistance upon nucleolar and fibrillarin reduction likely works through translational control, since mutants that diminish translation trigger similar pathogen resistance, and fibrillarin knockdown only modestly improves survival of such mutants, suggesting overlapping mechanisms. Furthermore, we observe reduced levels of mature rRNA and translation upon *fib-1* RNAi, adding more evidence that FIB-1 might orchestrate infection resistance via translational control. Surprisingly, *fib-1* RNAi had little overt effect on global proteasome or autophagy activity, although it is conceivable that it could still affect these processes in a tissue or stage specific manner. While elegant studies have previously shown an induction of the immune response genes by reduced translation[8], this work significantly extends these observations to demonstrate that this process broadly confers pathogen resistance. Since the nucleolus is involved in multiple cellular processes, however, it remains to be seen whether other nucleolar-regulated activities also influence immune function.

Fibrillarin is a highly conserved protein with similar structure and function in diverse species. Our study reveals a novel evolutionarily conserved function of fibrillarin in regulating innate immunity in mammals. Human epithelial cells and macrophages show a downregulation of fibrillarin after 24 h of infection, corroborating the results obtained in *C. elegans* and suggesting that reduction in fibrillarin levels as an ancestrally conserved host-defense response toward infection. Depletion of fibrillarin dampened the secretion of pro-inflammatory cytokines and increased anti-inflammatory cytokines upon *S. aureus* infection, which also correlated with diminished cell death. Similarly, fibrillarin knockdown in *C. elegans* stimulated *irg-1* expression but not a general transcriptional inflammatory response. This is consistent with the Damage Framework Model of Casadevall and Pirofski, which suggests that the severity of infection is guided by inflammation[43]. Inflammatory response to infection is required to defend against infection. However, overt inflammation makes the host susceptible to infection as a result of collateral damage to cells and tissues caused by the inflammatory cytokines. Therefore,

pathogen resistance mechanisms encompass negative regulators of inflammation, which are required to elicit optimal host anti-microbial response. Moreover, fusion of the pathogen containing phagosome with lysosomes plays a vital role in containing infection, and inflammatory stimuli are also known to affect the process of phagosome maturation[44]. Consistently, we also observed an increased co-localization of intracellular bacteria with lysosomes in cells bearing fibrillarin knockdown. This intriguing observation points toward a possible role of fibrillarin in mediating lysosomal biogenesis or acidification, which needs to be further investigated. Whether the observed reduction in pro-inflammatory cytokine generation and increased cell survival is a result of accelerated phagosome maturation, translation regulation, or other processes affected by fibrillarin remains to be seen.

Our study opens multiple avenues to fundamentally explore the role of the nucleolus and fibrillarin in fighting pathogenic bacterial infections, and raises several important questions. What triggers fibrillarin downregulation upon infection? What is the mechanism by which this confers resistance? Might specific methylation sites in rRNA affect immune function? What are the dynamics of nucleolar and downstream processes driving pathogen resistance? With the current rise in antibiotic and multidrug resistant bacteria, future efforts toward the discovery of novel intrinsic cellular factors that hinder bacterial growth and improve host resistance may prove crucial. Conceivably, fibrillarin and related molecules could be used as vital targets for drug screens combating bacterial infections in mammals.

## Methods

**C. elegans strains**. Worm strains were maintained at 20 °C following standard procedures[45]. N2 (wild type), ncl-1(e1865), ncl-1(e1942), daf-16(mu86), hlh-30 (tm1978), adIs2122(lgg-1::gfp; rol-6(su1006)), cguIs001 (FIB-1::GFP)[46], ife-2 (ok306), ifg-1(cxTi9279), pmk-1(km25), agIs17 (myo-2p::mCherry + irg-1p::GFP).

**Killing assay plate preparation**. *S. aureus* (strain MW2-WT), *E. faecalis* (strain ATCC 29212), and *P. aeruginosa* (strain PA14) were used for infection in *C. elegans*. *S. aureus* was grown in tryptic soy broth medium. *E. faecalis* was grown in brain heart infusion (BHI) medium. *P. aeruginosa* was grown in lysogeny broth. Twenty microliters of saturated overnight bacterial cultures were spread on tryptic soy agar (TSA) with 10 µg/mL nalidixic acid (Sigma, NAL) for *S. aureus*, BHI with 10 µg/mL NAL for *E. faecalis*, and modified NGM, 0.35% peptone, for *P. aeruginosa*. The plates were then incubated at 37 °C overnight.

**C. elegans killing assay**. Age synchronized young adults were transferred to killing assay plates and the survival assay was carried out at 25 °C. For each condition, three technical replicates were set up with 20 worms on each plate. Scoring was performed every 12 h for *S. aureus* and *P. aeruginosa* and every 24 h for *E. faecalis*. Worms were scored as dead if they failed to respond to gentle touch with a worm pick. Animals that crawled off the plate or had vulval explosion were censored.

**qRT-PCR**. Quantitative reverse transcription polymerase chain reaction (qRT-PCR) was performed to measure RNA transcript levels. Age synchronized young adults were washed three times in M9 buffer and then transferred to TSA plates with 10 µg/mL nalidixic acid (Sigma, NAL) carrying either *S. aureus* or heat-killed OP50 at 25 °C. At the indicated times, animals were harvested and washed twice with M9 before lysis. Worms were lysed in QIAzol Lysis Reagent (Qiagen). RNA was isolated using RNeasy Mini kit (Qiagen). Complementary DNA synthesis was performed using iScript cDNA Synthesis Kit (Bio-Rad). Experiments were performed according to manufacturer's instructions. Quantitative PCR was performed on a ViiA 7 Real-Time PCR System (Applied Biosystems) using Power SYBR Green master mix (Applied Biosystems). All the experiments were performed three times independently and the results were normalized to *snb-1*. qRT-PCR primer sequences are given in Supplementary Table 1. The primer sequences for pre-rRNA species were adopt from a recent study[47].

**Western blotting**. For worm western blot, 50 animals were transferred to killing assay plates and incubated at 25 °C for 12 h. Thirty animals from each condition were suspended in Laemmli lysis buffer with 5% 2-mercaptoethanol (Sigma). For cells, 20 µg of protein were loaded on the gels. The samples were boiled at 95 °C and loaded on Bio-Rad Midi-PROTEAN precast gels. Proteins were separated by standard sodium dodecyl sulfate–polyacrylamide gel electrophoresis (SDS-PAGE) and transferred to nitrocellulose membranes. The membranes were then blocked

with 5% milk before probing with the following antibodies overnight: anti-fibrillarin (Novus Biologicals NB300-269, 1:1000), anti-Histone H3 (Abcam ab1791, 1:4000), anti-βActin (Abcam ab8224, 1:5000), anti-GAPDH (Santacruz sc47724, 1:2500), anti-puromycin (Millipore 12D10, 1:10000), anti-ubiquitin (Cell Signaling P4D1, 1:1000), anti-GFP (Takara 632381, 1:2000), anti-mouse HRP (ThermoFisher G-21040, 1:5000), anti-rabbit HRP (ThermoFisher G-21234, 1:5000). HRP means the antibodies are horseradish peroxidase conjugated. Uncropped versions of all the western blots referred to in the main text and figures are provided in Supplementary Figure 7.

**Imaging and quantification**. Differential interference contrast (DIC) microscopy was used to perform all the nucleolar imaging. Hypodermal cells of age-matched day 1 adults were imaged using ×100 magnification with Olympus IX81. Freehand tool software from Fiji was used for nucleolar area quantification. FIB-1::GFP, HLH-30::GFP, and N2 + P$_{irg-1}$GFP worms were imaged using Olympus IX81 and Axio Imager Z1 (Zeiss).

**Puromycin incorporation assay**. Worms treated with RNAi as described were incubated in S-basal medium supplemented with 0.5 mg/mL puromycin and OP50 as food source for 3 h at 20 °C with gentle shaking. For the cycloheximide control, 2 mg/mL of the chemical was used. Thirty animals per condition were harvested for western blot detection of incorporated puromycin. The experiment was performed three times independently.

**rRNA analysis**. For infected samples, worms were treated the same as described for killing assay for 12 h. Hundred animals were harvested for each sample. Worms were lysed and total RNA was obtained following the same protocol as performed for qRT-PCR. Total RNA samples, extracted from equal number of worms were analyzed on Agilent 2200 TapeStation System following the High Sensitivity RNA ScreenTape System protocol (Agilent). The experiment was performed three times independently.

**Mammalian cell culture**. Human epithelial cell lines, HeLa was obtained from ATCC. The cells were cultured in Dulbecco's modified Eagle's medium supplemented with 10% fetal calf serum (FCS). The cells were maintained at 37 °C with 5% CO$_2$ in a humid atmosphere. The cells were tested for mycoplasma contamination.

Bone marrow-derived macrophages were prepared from 8–12 weeks old female C57BL/6J mice maintained and bred in the animal facility of Center for Molecular Medicine, University of Cologne, Germany. Mice were killed by cervical dislocation and bone marrows from the femurs were flushed using RPMI medium. The flushed cells were centrifuged and re-suspended in RPMI containing 10% FBS. Cells were seeded in culture dishes and allowed to differentiate into macrophages in medium supplemented with 20% L929 cell-culture supernatant for 7 days. Non-adherent cells were removed on days 2 and 4, and adherent macrophages were used from day 7 onwards. Mouse experiments were performed according to the guidelines and approval of LANUV (Landesamt für Natur, Umwelt und Verbraucherschutz Nordrhein-Westfalen (State Agency for Nature, Environment and Consumer Protection North Rhine-Westphalia)).

THP1 monocytes were obtained from ATCC. The cells were tested for mycoplasma contamination. The cells were maintained in RMPI 1640 media, supplemented with 10% FCS. For differentiation of THP1 monocytes into macrophages, phorbol 12-myristate 13-acetate (PMA, Sigma, P8139) was used. Briefly, THP1 derived monocytes were incubated in 10% FCS RPMI 1640 media supplemented with 25 ng/mL of PMA for 24 h. The differentiated cells were used for infection.

**Fibrillarin knockdown and overexpression**. siGenome siRNA for fibrillarin was obtained from Dharmacon (GE Healthcare Life Sciences). The cells were treated with 100 nM of siRNA 48 h before infection using Dharmafect-2 (GE Healthcare Life Sciences) according to the manufacturer's protocol. For mammalian plasmid transfection, plasmid carrying human fibrillarin fused to EGFP, cloned under CMV promoter (p-EGFP C1) was procured from Addgene (catalog no. 26673). The empty vector only carrying the EGFP was used as the control. HeLa cells were transfected with the plasmids using Lipofectamine 3000 (ThermoFisher Scientific) following the manufacturer's protocol. Transfected cells were assayed for fibrillarin and GFP expression 48 h post transfection and were then used for infection.

**Infection in mammalian cells**. A late logarithmic phase grown *S. aureus* (MW2), *E. faecalis* (ATCC 29212), *S. Typhimurium* (SL 1344), and *L. monocytogenes* (EGDe) were used at MOI 50 and MOI 10 for HeLa and macrophages, respectively. The cells were transfected with fibrillarin siRNA or overexpression plasmid for 48 h followed by infection. The infected cells were incubated for 10 min at room temperature, followed by an incubation for 30 min at 37 °C with 5% CO$_2$ in a humid atmosphere. After 30 min, extracellular bacteria were removed and cells were incubated for 2 h in medium containing 50 μg/mL gentamicin and then were washed and subsequently cultured in medium containing less gentamicin

(10 μg/mL). At desired time points cells were collected for western blot analysis using RIPA lysis buffer.

**Cell viability assay**. Cell viability was measured using a LDH Cytotoxicity Assay Kit (CytoTox 96 non-radioactive cytotoxicity assay; Promega). Released LDH was measured according to the manufacturer's protocol. The percentage of cell death was calculated using the formula: % Cell death = experimental release/maximum release × 100. Trypan Blue method was performed by treating cells with trypsin at different time points post infection and counting viable cells using standard Trypan Blue dye exclusion assay.

**Gentamycin protection assay**. After *S. aureus* infection, the cells were washed three times with sterile PBS and lysed with 0.3% Triton X-100 in PBS for 5 min at room temperature. Several dilutions of the lysate were plated on BHI plates and incubated overnight at 37 °C. The following day, *S. aureus* CFU were counted.

**Immunocytochemistry**. HeLa cells were infected with GFP expressing *S. aureus* as per the above-mentioned protocol. Twenty-four hours post infection, the cells were incubated with 250 nM lysotracker deep red (Invitrogen) for 15 min at 37 °C with 5% CO$_2$ in a humid atmosphere. Then the cells were washed with warm PBS and fixed at room temperature for 15 min in 4% paraformaldehyde. Fixed cells were washed three times with PBS and mounted on slides with ProLong Gold mounting medium containing DAPI (ThermoFisher Scientific). Images were acquired with a ×60 oil PlanApo objective numerical aperture 1.4 at room temperature on an Olympus IX81 inverted confocal microscope equipped with PMT detectors for imaging. Olympus Fluoview -10 ASW 4.2 software was used for acquisition and calculating Pearson's correlation.

**ELISA**. The supernatants from infected and uninfected cells were collected and snap frozen. ELISA was performed to gauge the levels of IL-6, IL-8, and IL-10 using DUOSet ELISA kits from R&D Biosystems, following the manufacturer's protocol.

**Blinding of experiments**. All the *C. elegans* killing assays were performed in a blinded manner. For blinding, the bacterial strain and *C. elegans* strain names were concealed during scoring, analyzing, and plotting the data. Nucleolar imaging and quantification were also performed with concealed strain names. For blinding mammalian cell-culture experiments, one experimenter performed the siRNA transfections, and the experiment was blinded henceforth. The infections and other biochemical assays were carried out by the other experimenter.

## Data availability
The authors declare that all the data and the methods used in this study are available within this article, its Supplementary Information files, the peer-review file, or are available from the corresponding author on request.

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

## Acknowledgements

We thank *C. elegans* Genetic Center (CGC), University of Minnesota for providing *C. elegans* strains, Saray Gutierrez for help with ELISA experiments, and Jennifer Klimek for help with bacterial clearance experiments. This work was supported by the Max Planck Society, CECAD/Deutsche Forschungsgemeinschaft (DFG), and Cologne Graduate School of Ageing Research/DFG doctoral scholarship (V.T.).

## Author contributions

V.T., C.K., P.M., N.R., and A.A. designed the study. V.T., C.K., P.M., R.G., and N.R. performed experiments and data analysis. V.T., P.M., N.R., and A.A. wrote the manuscript.

## Additional information

**Competing interests:** The authors declare no competing interests.

