## [Peer Review File · Nature Communications]

Reviewers' comments:

Reviewer #1 (Remarks to the Author):

In this manuscript, Tiku et al. are reporting in worm and human cells a connection between pathogen infection, the expression of the methyltransferase Fibrillarlin (FBL), and nucleolar size.

Specifically, they show both in *C. elegans* and human cells that the size of the nucleolus and the levels of expression of FBL are reduced upon infection. In *C. elegans*, they further show that FBL knockdown increases resistance to bacterial pathogens while enforced FBL expression (seen in *ncl1-1* mutants) makes worm more susceptible to infection. Apparently, FBL acts independently of major innate immunity mediators (epistasis analysis). In human cells, they also found that FBL KD prior to infection reduces inflammation and improves cell survival. Finally, they provide preliminary evidences that translation may be involved.

This is a very interesting manuscript and I really enjoyed reading it. There are only few cases of connections between ribosome biogenesis/nucleolar structure alterations and infection.

Fibrillarlin is an abundant nucleolar protein which plays multiple roles in cells: (1) As part of large precursor ribosomes, it is required for essential pre-rRNA processing (cleavage) reactions, (2) As part of numerous small ribonucleoprotein particles (snoRNPs), it is involved in non-essential 2'-O-methylation of precursor ribosomal RNAs, and (3) It has also been involved rDNA histone methylation (Although this later function is more loosely defined and has not been confirmed).

Fibrillarlin can thus both modulate the amount of ribosome produced (quantitative aspect) and the post-transcriptional landscape of the rRNAs (qualitative aspect).

Although one cannot exclude that FBL plays additional functions in cells (possibly with other partners, in other complexes etc.), based on its known functions, it seems quite logical to research a connection between infection and ribosome biogenesis/function. And this is where the authors should improve their work before it can be accepted in Nature Communications.

What I would like to see is:

- (1) Test how the reduced/increased expression of FBL impacts ribosome production, i.e. establish mature rRNA levels and pre-rRNA processing.
- (2) Test how the reduced/increased expression of FBL impacts ribosomal RNA 2'-O-methylation, i.e. perform quantitative RiboMethSeq analysis.
- (3) Test how global translation is affected upon reduced/increased expression of FBL, e.g. puromycin incorporation, polysome profiles etc - this later point is less important, as it is partly redundant with the ribosome production analysis.

To me, the most important is to monitor the effects of alterations of FBL expression on ribosome biogenesis and rRNA ribose methylation.

With this, the authors will be able to tell whether it's just a case of the amount of ribosomes available in the cells (if it is, then they should see exactly the same effect with any other ribosomal assembly factors, there are several hundreds of them...) or whether different ribosomes are produced (more or less methylated) which would suggest that differential translation may be involved. Interestingly, in a recent study (doi:10.1038/s41598-017-09734-9), we have shown that upon a controlled and progressive depletion of FBL specific sites appeared more vulnerable than others and, surprisingly, that these correspond largely to sites which are naturally hypomodified (i.e. not made on all ribosomes). It would be interesting to see if similar sites are affected in the context of an infection.

Other comments:

-HeLa cells are not ideal here because they do not express p53 physiologically and FBL' expression is regulated by p53
-in the epistasis analysis, I think FBL acts upstream since it is dominant
-typo 'cycloheximide' in Supplementary Figure 5 panel A

This report was prepared by Denis Lafontaine

Reviewer #2 (Remarks to the Author):

In "Nucleolar Fibrillarin is an evolutionarily conserved regulator of bacterial pathogen resistance," the authors report that this factor is vital in controlling susceptibility to infection in *C. elegans* as well as in mammalian cells in a well-written manuscript. They show that knockdown of Fibrillarin enhances resistance while higher levels increase susceptibility. Infection reduces nucleolar size and Fibrillarin levels, suggesting that this is a purposeful mechanism of protection. In trying to understand the mechanism, they show that Fibrillarin improves the resistance of most infection sensitive mutants. However, a connection to translation is found – Fibrillarin reduction induces the translation suppression reporter *irg-1::GFP*. This is not surprising, since Fibrillarin is involved in rRNA maturation and ribosome biogenesis and therefore is needed for translation. It is known that defects in translation activate pathogen response and increase resistance (Dunbar et al. *Cell Host Microbe*, 2012, Melo et al. *Cell* 2013, lessening the novelty of these results.

- 1) The authors show that inhibiting/slowing down translation by other means such as knocking down *ifg-1* and *ife-2* and adding cycloheximide also increases survival. This has been shown previously (Dunbar et al. *Cell Host Microbe*, 2012, Melo et al. *Cell* 2013). Because knocking down Fibrillarin also slows down translation, the authors' results are not that surprising. However, the authors fail to put them in the broader context that it has been shown that animals undergoing infection experience high levels of proteotoxic stress. Getting rid of systems that deal with damaged proteins such as autophagy, the proteasome, and chaperones increases proteotoxic stress and decreases survival. Reducing the amount of protein that the folding/repair machinery has to deal with by inhibiting/slowing down translation, decreases proteotoxic stress and increases survival. More discussion on these points is recommended.
- 2) The authors should test whether the induction of *irg-1* is dependent on ZIP-2 or not. The block in translation caused by loss of Fibrillarin could very well be activating protection via ZIP-2/CEBP-2. On the other hand, *irg-1* induction by ZIP-2-independent mechanisms have been shown (Dunbar et al. *Cell Host Microbe*, 2012).
- 3) Based on the author's previous work, Fibrillarin also causes an increase in longevity. Is this also due to a lessening of proteotoxic stress in the system? Some discussion would be desirable.
- 4) Sup Figure 1G is not necessary. It has been shown in previous work that killing by *S. aureus* requires live bacteria. What would be more informative is looking at the lifespan of wild type vs. Fibrillarian knock-down on dead *S. aureus* vs. live *S. aureus* to see the degree to which Fibrillarian effects lifespan on pathogenic vs. non-pathogenic conditions.

Reviewer #1:

“Although one cannot exclude that FBL plays additional functions in cells (possibly with other partners, in other complexes etc.), based on its known functions, it seems quite logical to research a connection between infection and ribosome biogenesis/function. And this is where the authors should improve their work before it can be accepted in Nature Communications.”

Comment 1: Test how the reduced/increased expression of FBL impacts ribosome production, i.e. establish mature rRNA levels and pre-rRNA processing.

Response: *In order to address the concerns raised by the reviewer, we probed the effect of Fib-1 knockdown on the levels of pre-rRNA by qRT-PCR (Suppl. Fig 5c) and mature rRNA by Bio-analyzer (Figure 4f) (see below). We observed that Fib-1 RNAi leads to an increase in the abundance of pre-rRNA, and a concurrent decrease in mature rRNA levels. This strongly suggests an accumulation of the precursor and a diminution of the products of Fib-1 processing caused by knockdown. Furthermore, we observed that reduced mature rRNA levels could also be seen under infection (where Fib-1 levels are reduced): just as with Fib-1 knock down, S. aureus infection of C. elegans reduced the levels of mature rRNA (Figure 2f) (see below). Together, these results suggest that a reduction in Fib-1 levels exerts its effect by interfering with rRNA maturation and that infection on its own brings down the mature levels of rRNA.*

Illustration for the pre-rRNA qPCR primers

Comment 2: Test how the reduced/increased expression of FBL impacts ribosomal RNA 2'-O-methylation, i.e. perform quantitative RiboMethSeq analysis.

Response: *The reviewer has raised an interesting point here and we agree that it will be useful to elucidate how expression levels of Fibrillarlin impact ribosomal 2'-O-methylation at different residues. However, since there exists only a partial map of C. elegans rRNA modifications, seeking this information would require characterizing the complete RiboMethSeq, seeing if this is quantifiably modified upon fib-1 knockdown, and if so, whether this maps to infection resistance. This would require a much more in depth experimental design and data analysis and lies beyond the scope of this paper. This is something that we are currently pursuing in another project.*

Comment 3: Test how global translation is affected upon reduced/increased expression of FBL, e.g. puromycin incorporation, polysome profiles etc - this later point is less important, as it is partly redundant with the ribosome production analysis.

Response: *This is an excellent suggestion. In order to address this query, we performed Puromycin incorporation assay, as suggested by the reviewer. As expected, we observed that knocking down Fibrillarlin reduced the amount of puromycin incorporation, indicative of reduced translation (Figure 4G,H) (See below). This observation, coupled to the fact that a reduction of Fib-1 also reduces mature rRNA levels, strongly suggests that Fib-1 modulates translation globally by influencing ribosome biogenesis.*

Comment 4: With this, the authors will be able to tell whether it's just a case of the amount of ribosomes available in the cells (if it is, then they should see exactly the same effect with any other ribosomal assembly factors, there are several hundreds of them...) or whether different ribosomes are produced (more or less methylated) which would suggest that differential translation may be involved. Interestingly, in a recent study (doi:10.1038/s41598-017-09734-9), we have shown that upon a controlled and progressive depletion of FBL specific sites appeared more vulnerable than others and, surprisingly, that these correspond largely to sites which are naturally hypomodified (i.e. not made on all ribosomes). It would be interesting to see if similar sites are affected in the context of an infection.

Response: *Since mature rRNA levels are reduced in infected animals, it is likely that there are less ribosomes available in the cells. However, at this point, we cannot conclude whether ribosomal modification is implicated. As suggested above, we are currently trying to investigate rRNA modification changes using RiboMethSeq and other methodologies. The experiments are included in a separate project specialized in rRNA maturation and ribosomal modification processes.*

Comment 5: HeLa cells are not ideal here because they do not express p53 physiologically and FBL' expression is regulated by p53

Response: *Although HeLa cells have been widely used for studying S. aureus infection and host response, we understand the reviewer's concern. We have now performed all the experiments on bone marrow derived mouse macrophages (BMDMs). The results are consistent with the observations made in HeLa cells. Reduction of FBL facilitates intracellular bacterial clearance in infected BMDMs as measured by colony forming units (CFU) assay (Figure 5G) (see below). Furthermore, host cell survival was improved (cytotoxicity as measured by LDH release assay) in cells treated with FBL siRNA prior to infection (Figure 5F) (see below). Additionally, we also observed the anti-inflammatory cytokine IL-10 to be induced upon FBL knockdown in BMDMs (Figure 5I) (see below). Taken together, these results from primary macrophages recapitulate the observations made in HeLa cells, that FBL reduction has beneficial effects on host cell survival during infection and enhances intracellular bacterial clearance.*

Comment 6: In the epistasis

analysis, I think FBL acts upstream since it is dominant

Response: *In classic epistasis analysis, the prevailing gene is placed downstream in a signaling pathway.*

Comment: Typo 'cycloheximide' in Supplementary Figure 5 panel A

Response: *Corrected*

Reviewer #2

Comment 1: The authors show that inhibiting/slowing down translation by other means such as knocking down ifg-1 and ife-2 and adding cycloheximide also increases survival. This has been shown previously (Dunbar et al. Cell Host Microbe, 2012, Melo et al. Cell 2013).

Response: *We thank the reviewer for this comment. In our read of these papers, the authors do not directly test survival upon translational knockdown. Dunbar et. al. show that P. aeruginosa toxin Tox-A induces translation inhibition with a consequent increase in ZIP-2 protein levels, which is responsible for turning on immunity genes including irg-1. Zip-2 itself has very mild effect on P. aeruginosa susceptibility. Melo et. al. show that inhibition of core cellular processes, including translation, induce immunity genes and toxic food aversion behavior.*

Although these papers link translation to irg-1 gene expression and aversion behavior, they do not establish that diminished translation itself enhances survival. Our observation is novel in that it shows that fib-1 inhibition and translation reduction leads to dramatically increased survivorship upon pathogen challenge, and shows a critical physiologic consequence to reduced nucleolar function and translation reduction. More importantly, our work extends this concept to mammalian cells, including dedicated immune cells such as macrophages, showing the evolutionary conservation of this phenomenon.

Comment 2: Because knocking down Fibrillarlin also slows down translation, the authors' results are not that surprising. However, the authors fail to put them in the broader context that it has been shown that animals undergoing infection experience high levels of proteotoxic stress. Getting rid of systems that deal with damaged proteins such as autophagy, the proteasome, and chaperones increases proteotoxic stress and decreases survival. Reducing the amount of protein that the folding/repair machinery has to deal with by inhibiting/slowing down translation, decreases proteotoxic stress and increases survival. More discussion on these points is recommended.

Response: *The reviewer raised interesting points about the potential effect on proteostasis mechanisms upon Fibrillarlin reduction. To investigate these queries, we started by checking the ubiquitin proteasome system (UPS). We compared the overall levels of ubiquitinated proteins in C. elegans infected with S. aureus with or without fib-1 knockdown. In either uninfected or infected C. elegans, we could not observe a change in total levels of ubiquitinated proteins (Supplementary Figure 4D) (see below). rpn-6, a component of the UPS, served as a positive control and knock down of this gene clearly showed an accumulation of ubiquitinated proteins suggesting a reduction in function of the UPS. Further, we went on to analyze if the autophagic machinery is altered by fib-1 knockdown. In an assay determining the levels autophagosomal membrane protein LC3/LGG-1, we could observe that fib-1 RNAi alone does not affect autophagic flux under normal and infection conditions (Supplementary figure 4C) (See below). Starvation and infection, which are known to induce autophagy, showed a visible increase of free GFP/total GFP, indicating increased flux, and validating our assay (See below).*

Comment 3: The authors should test whether the induction of irg-1 is dependent on ZIP-2 or not. The block in translation caused by loss of Fibrillarlin could very well be activating protection via ZIP-2/CEBP-2. On the other hand, irg-1 induction by ZIP-2-independent mechanisms have been shown (Dunbar et al. Cell Host Microbe, 2012).

Response: We thank the reviewer for this comment. We measured *irg-1* induction in the *zip-2* deletion mutant. Knocking out *zip-2* abolished induction of *irg-1* upon *fib-1* reduction by RNAi, suggesting induction of *irg-1* indeed is dependent on *zip-2*. We also tested if *zip-2* is required for infection resistance conferred by *fib-1* reduction. Interestingly, *fib-1* RNAi leads to similar resistance in both wildtype and *zip-2* mutant upon infection by two different pathogenic bacteria (*Staphylococcus aureus* and *Pseudomonas aeruginosa*). Based on these observations, we conclude that induction of *irg-1* by *fib-1* reduction is dependent on *zip-2*, but the resistance phenotype is *zip-2* independent (see below).

Comment 4: Based on the author's previous work, Fibrillarlin also causes an increase in longevity. Is this also due to a lessening of proteotoxic stress in the system? Some discussion would be desirable.

Response: We have done assays to check proteotoxicity and autophagic flux and *fib-1* knockdown does not obviously alter proteotoxic stress or autophagic flux (please see above in response to Comment 2, the second reviewer).

Comment 5: Sup Figure 1G is not necessary. It has been shown in previous work that killing by *S. aureus* requires live bacteria. What would be more informative is looking

at the lifespan of wild type vs. Fibrillarian knock-down on dead *S. aureus* vs. live *S. aureus* to see the degree to which Fibrillarian effects lifespan on pathogenic vs. non-pathogenic conditions.

Response: *Thank you. We have removed the mentioned figure.*

Reviewers' comments:

Reviewer #1 (Remarks to the Author):

The authors have adequately addressed my comments, for most of them. It's a shame they could not perform RiboMethSeq in the timeframe of the revision, as this would have provided useful mechanistic insights. But I am glad to hear they are pursuing this.

Reviewer #2 (Remarks to the Author):

Overall, the authors have done a very nice job of responding to my previous comments. Unfortunately, I have one more major concern that I missed in my initial review. In responding to my Comment #1 about the previous work (Dunbar et al. Cell Host Microbe, 2012, Melo et al. Cell 2013) the authors mention that this previous work never examined survival but rather looked at aversion response and/or immune gene regulation. That is correct. The problem shown by Melo et al. is that translation inhibition causes a strong aversion response – the worms actively avoid the pathogen. A stronger/weaker aversion response has been shown to affect overall worm survival and has caused problems/controversy in the *C. elegans* pathogenesis field in the past (For example, see these Science papers – Styer et al. Science. 2008 and Reddy et al. Science. 2009). To avoid this problem, one must do the pathogen killing assays on full-lawn plates when working with strains/RNAi knock-downs that might have altered behavior such that the worms cannot avoid exposure to the pathogen. From my examination of the text, I cannot tell if the assays were done in this manner or in the more traditional manner of just spotting a lawn on the center of the plate and leaving a clear area around it. (One could also do liquid killing assays, which also forces continuous pathogen exposure without the problem of the worms crawling up the sides of the plates.) My final request of the authors is that this control be done (if not done already) and added to the manuscript. I apologize for not catching this during the first round of review.

Response to Reviewers

Reviewer #1 :

“The authors have adequately addressed my comments, for most of them. It's a shame they could not perform RiboMethSeq in the timeframe of the revision, as this would have provided useful mechanistic insights. But I am glad to hear they are pursuing this”

Response:

We greatly appreciate the gesture extended by the journal of furthering the revision timescale allowing for Ribo Meth Seq analysis. We completely agree with the editor and the reviewer about the Ribo-met Seq providing useful insights into the mechanism of FIB-1 in immunity. However, as noted before the rRNA methylation landscape in C. elegans is poorly defined and needs extensive analysis and characterization before a method like Ribo Meth Seq could be employed to make further interpretations. A complete characterization of rRNA methylation sites in C. elegans, followed by the investigation of the effects of fib-1 knockdown on these methylation sites and finally mapping which of the modified sites correspond to infection resistance is a whole new study and requires extensive amount of time and effort. This kind of analysis is beyond the scope of this paper. As mentioned earlier, this study has been taken up as a whole new project in the lab and is being pursued further. Therefore, we request the editor to accept the manuscript in its current state.

Reviewer #2

Overall, the authors have done a very nice job of responding to my previous comments. Unfortunately, I have one more major concern that I missed in my initial review. In responding to my Comment #1 about the previous work (Dunbar et al. Cell Host Microbe, 2012, Melo et al. Cell 2013) the authors mention that this previous work never examined survival but rather looked at aversion response and/or immune gene regulation. That is correct. The problem shown by Melo et al. is that translation inhibition causes a strong aversion response – the worms actively avoid the pathogen. A stronger/weaker aversion response has been shown to affect overall worm survival and has caused problems/controversy in the C. elegans pathogenesis field in the past (For example, see these Science papers – Styer et al. Science. 2008 and Reddy et al. Science. 2009). To avoid this problem, one must do the pathogen killing assays on full-lawn plates when working with strains/RNAi knock-downs that might have altered behavior such that the worms cannot avoid exposure to the pathogen. From my examination of the text, I cannot tell if the assays were done in this manner or in the more traditional manner of just spotting a lawn on the center of the plate and leaving a clear area around it. (One could also do liquid killing assays, which also forces continuous pathogen exposure without the problem of the worms crawling up the sides of the plates.) My final request of the authors is that this control be done (if not done already) and added to the manuscript. I apologize for not catching this during the first round of review.

Response:

The reviewer's concern is valid and we have performed further experiments to address the concern. As suggested by the reviewer, we performed experiments where both the control and *Fib-1* RNAi worms were put on bacterial lawns of *S. aureus* and *P. aeruginosa* (spread all over the plate). In accord with our findings, it was observed that the worms bearing *fib-1* RNAi out lived the control worms (added in Supplementary figure 1G and 1H). Previously we also confirmed no difference in pharyngeal pumping rates between WT and *fib-1* RNAi worms (shown in Supplementary figure 1F). Together these data demonstrate the fact that *fib-1* knock down does not lead to any food behavioral perturbation which might be causing the observed resistance to infection.

Control or *fib-1* KD bearing *C.elegance* were transferred to NGM plates covered entirely in either *S.aureus* or *P.aeruginosa*. The worms were followed for their survival on the pathogens.

REVIEWERS' COMMENTS:

Reviewer #2 (Remarks to the Author):

Now that experiments to control for possible food aversion effects are included, all my concerns are addressed.

Danielle Garsin